# Sexual behavior is linked to changes in gut microbiome and systemic inflammation that lead to HIV-1 infection in men who have sex with men

Huang Lin [1,9,10], Yue Chen [2,10], Grace Abror-Lacks[3], Meaghan Price[3], Alison Morris[4], Jing Sun [5], Frank Palella[6], Kara W. Chew [7], Todd T. Brown[8], Charles R. Rinaldo[2,11] ✉ & Shyamal D. Peddada [1,11] ✉

Pathogenic changes in gut microbial composition precede the onset of HIV-1 infection in men who have sex with men (MSM). This process is associated with increased levels of systemic inflammatory biomarkers and risk for AIDS development. Using mediation analysis framework, in this report we link the effects of unprotected receptive intercourse among MSM prior to primary HIV-1 infection to higher levels of proinflammatory cytokines sCD14 and sCD163 in plasma and a significant decrease in the abundance of *A. muciniphila, B. caccae, B. fragilis, B. uniformis, Bacteroides* spp*., Butyricimonas* spp., and *Odoribacter* spp., and a potential increase in the abundance of *Dehalobacterium* spp. and *Methanobrevibacter* spp. in stools of MSM with the highest number of sexual partners. These differences in microbiota, together with a reduction in the pairwise correlations among commensal and short-chain fatty acid-producing bacteria with a number of sexual partners, support an increase in gut dysbiosis with the number of sexual partners. These results demonstrate the interconnectedness of sexual behavior, immune response, and microbiota composition, notably among MSM participating in high-risk sexual behaviors.

Since the beginning of the HIV-1 epidemic in the USA, sexual contact among men who have sex with men (MSM) has been the primary route of HIV-1 transmission in the US[1]. Sexual transmission currently accounts for more than 90% of new HIV-1 infections in the US, with 68% transmitted through sexual contact between MSM[2,3]. Anal intercourse is the riskiest sexual practice for acquiring or transmitting HIV-1, with the receptive partner at higher risk than the insertive partner[4–8], particularly a condomless receptive anal intercourse partner[9,10]. Advances in our understanding of HIV-1 pathogenesis have revealed the critical role of cytokines and the gut microbiome in HIV-1 transmission and disease progression[11]. Gut microbiome dysbiosis can increase local inflammatory cytokine production, thereby activating CD4[+] T cells and increasing HIV-1 co-receptor CCR5 expression on CD4[+] T cells in gut-associated lymphoid tissues (GALT), the cells preferentially infected by HIV-1[12–14]. The presence of activated CD4[+] T cells in the rectum and anus makes MSM who practice unprotected anal receptive sexual intercourse particularly susceptible to HIV-1 infection. Moreover, inflammation and immune activation of GALT lead to the translocation of gut bacteria and bacterial products into the blood circulation, resulting in systemic immune activation that fuels the dissemination of HIV-1 when initial viral exposure occurs in local GALT[4,15]. Cytokines, as key mediators of immune responses, can modulate the activation and function of immune cells, including CD4[+] T cells that are the primary target of HIV-1. Elevated systemic levels of proinflammatory cytokines contribute to a heightened state of immune activation, rendering individuals more susceptible to HIV-1 infection[16–18].

[1]Biostatistics and Computational Biology, National Institute of Environmental Health Sciences (NIH), Research Triangle Park, NC, USA. [2]Division of Infectious Diseases, Department of Medicine, University of Pittsburgh, Pittsburgh, PA, USA. [3]Department of Infectious Diseases and Microbiology, School of Public Health, University of Pittsburgh, Pittsburgh, PA, USA. [4]Division of Pulmonary, Allergy and Critical Care Medicine, Department of Medicine, University of Pittsburgh School of Medicine, Pittsburgh, PA, USA. [5]Department of Epidemiology, The Johns Hopkins Bloomberg School of Public Health, Baltimore, MD, USA. [6]Department of Medicine, Feinberg School of Medicine, Northwestern University, Chicago, IL, USA. [7]School of Medicine, University of California at Los Angeles, Los Angeles, CA, USA. [8]Department of Medicine, Johns Hopkins School of Medicine, Baltimore, MD, USA. [9]Present address: Department of Epidemiology and Biostatistics, University of Maryland, College Park, MD, USA. [10]These authors contributed equally: Huang Lin, Yue Chen. [11]These authors jointly supervised this work: Charles R. Rinaldo, Shyamal D. Peddada. ✉e-mail: rinaldo@pitt.edu; peddada@nih.gov

**Table 1 | Summary of demographic, clinical, and behavior features**

|  |  | **N = 243** |
|---|---|---|
| Age, (mean (SD)) |  | 40.70 (16.11) |
| Race, (n (%)) |  |  |
|  | White | 230 (95) |
|  | Black | 8 (3) |
|  | Others | 4 (2) |
|  | Missing | 1 (0) |
| Education, (n (%)) |  |  |
|  | Postgrad | 95 (39) |
|  | Undergrad | 58 (24) |
|  | No degree | 88 (36) |
|  | Missing | 2 (1) |
| Smoking status, (n (%)) |  |  |
|  | Never | 97 (40) |
|  | Former | 42 (17) |
|  | Current | 103 (42) |
|  | Missing | 1 (0) |
| Antibiotics usage, (n (%)) |  |  |
|  | No | 128 (53) |
|  | Yes | 115 (47) |
| STI, (n (%)) |  |  |
|  | No | 203 (84) |
|  | Yes | 39 (16) |
|  | Missing | 1 (0) |
| Substance use, (n (%)) |  |  |
|  | No | 45 (19) |
|  | Yes | 197 (81) |
|  | Missing | 1 (0) |
| HBV, (n (%)) |  |  |
|  | Negative | 96 (40) |
|  | Resolved | 133 (55) |
|  | Positive | 8 (3) |
|  | Missing | 6 (2) |
| HCV, (n (%)) |  |  |
|  | Negative | 231 (95) |
|  | Positive | 6 (2.5) |
|  | Missing | 6 (2.5) |
| Sexual exposure group, (n (%)) |  |  |
|  | G1 | 63 (26) |
|  | G2 | 57 (23) |
|  | G3 | 86 (35) |
|  | G4 | 35 (14) |
|  | Missing | 2 (1) |

Compared to men who have sex with women (MSW), MSM—regardless of HIV-1 serostatus—have a Prevotella-rich fecal[19,20] and rectal mucosal[5] microbiome, as well as a rectal mucosal immune activation profile. We previously reported that high levels of plasma inflammatory cytokines and gut microbiome profiles characterized by increased Prevotella and decreased Bacteroides a few months before HIV-1 infection were significantly associated with subsequent, primary HIV-1 infection in over 100 MSM from the 1984–1985 Multicenter AIDS Cohort Study (MACS)

cohort[21]. Consistent with this finding, the odds of HIV-1 infection were highly positively correlated with the ratio of *Prevotellaceae* to *Bacteroideceae*. The altered microbiome in MSM prior to HIV-1 infection was confirmed by Fulcher et al.[22] in a study of 2014–2018 non-MACS HIV-1 research cohorts. However, to the best of our knowledge, the interactions among sexual behavior, biomarker levels including cytokines, and the gut microbiome with subsequent HIV-1 infection have not been investigated. Increasingly researchers are interested in understanding the mediating role of gut microbiome on various health outcomes and exposures. For example, ref. 23 demonstrated that the gut microbiome mediates the effects of diet on immune function.

Capitalizing on the valuable longitudinal data collected before and after HIV-1 infection in the MACS, in the current study we have comprehensively analyzed: (1) the associations and interactions among self-reported sexual behaviors associated with increased risk for HIV-1 infection (number of partners with whom a participant had receptive anal intercourse, the primary exposure variable in this study), systemic inflammatory cytokine levels, and gut microbiome before HIV-1 infection in MSM participants with or without subsequent HIV-1 infection (the outcome variable in this study), and (2) whether changes in these biomarker levels can be considered as mediators through which sexual behavior impacts HIV-1 infection. Our results show that collectively rather than individually the biomarkers soluble CD14 (sCD14) and soluble scavenger receptor (sCD163), and the microbial species *Akkermansia muciniphila, Bacteroides caccae, Bacteroides fragilis, Bacteroides uniformis, Bacteroides* spp., *Butyricimonas* spp., *Dehalobacterium* spp., *Methanobrevibacter* spp., and *Odoribacter* spp., mediate the effects of sexual behavior on HIV-1 infection. Our research provides important insights into the role of the complex interplay among sexual behavior, gut microbiome, and systemic inflammation in HIV-1 transmission in MSM.

## Results
### Study participants and background data
We analyzed plasma cytokine levels, and microbiome data of the same study participants ($N = 241$) from our previous investigation[21], considering the participants' reported sexual activity. The cohort was exclusively MSM, and the average age (±standard deviation [SD]) of study participants was 41 ± 16 years (Table 1). The majority of these 1984–1985 MACS participants were white (95%) and achieved undergraduate or post-graduate college degrees (63%). The group included a higher number of non-smokers, with 57% either having never smoked or were former smokers, and current smokers comprising 42% of the total. A higher percentage of participants (56%) were categorized as heavy drinkers (consumption of three or more drinks per day at least once a month), as opposed to low drinkers (42%, defined as consumption of less than two drinks per day no more than once a month). Prevalent oral bacterial antibiotic use was reported by 53% of the cohort. Approximately 16% of participants reported having had at least one sexually transmitted infection (STI), including syphilis, non-specific/nongonococcal urethritis, chlamydia, herpes, and gonorrhea. A notable proportion (81%) reported past use of one or more substances including marijuana, poppers, cocaine, uppers, ecstasy, heroin, speedball, phencyclidine, downers, ethyl chloride, gamma-hydroxybutyrate, or other unspecified drugs. About 40% of the participants were hepatitis B virus (HBV) negative (anti-hepatitis B virus core (anti-HBc) negative and hepatitis B surface antigen (HBsAg) negative), 55% had resolved HBV infection (anti-HBc positive and HBsAg negative), and 3% were HBV positive at the time of the survey (HBsAg positive). About 95% of participants were hepatitis C virus antibody negative, 2.5% were positive, and the remaining 2.5% had missing test results. Participants were categorized into four ordered groups (G1 to G4) based on the number of partners with whom they engaged in receptive anal intercourse (see Methods for detailed definitions).

### Exploratory analyses
All the data of this study were collected in the early years of the AIDS epidemic before the availability of effective anti-HIV-1 therapy. We found that the HIV-1 infection rates monotonically increased with the number of

sexual partners with whom a participant had receptive anal intercourse during the previous 2 years ($p < 0.001$ Fig. 1). This finding suggests that the risk of HIV-1 infection increases with the number of partners with whom a man had receptive anal intercourse.

There were no statistically significant differences observed among the four sexual exposure groups, or between the HIV-1 infection status, with respect to mean ages, alcohol consumption status, education level, and smoking status (Fig. 2). However, recreational substance use was positively associated with the number of receptive anal intercourse partners ($p < 0.01$, Fig. 2a) and the risk of HIV-1 infection ($p = 0.02$, Fig. 2b). These findings

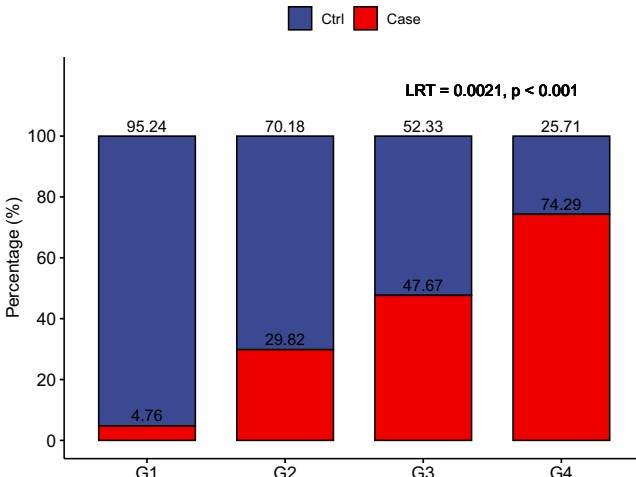

**Fig. 1 | HIV-1 infection monotonically increases with the number of receptive anal intercourse partners.** The $X$-axis categorizes participants ($N = 241$) into exposure groups, ranging from Group 1 (G1) through Group 4 (G4). The $Y$-axis represents the percentage distribution of the participants with subsequent HIV-1 infection (Case, depicted in red) and negative controls (Ctrl, depicted in blue). Combined, these percentages total 100%. As participants move from G1 to G4, indicative of an increase in the number of partners, there is a marked rise in the percentage of HIV infection. This increase is statistically significant supported by the likelihood ratio test (LRT) statistics of 0.0021 and a $p$-value (one-sided) of less than 0.001, as determined by the constrained linear mixed effects (CLME) test, suggesting a monotonic increasing trend.

suggest that substance use acts as a potential confounder between the sexual exposure groups and outcome variables. Antibiotic usage was positively associated with sexual exposure groups ($p = 0.03$, Fig. 2a) and with HIV-1 infection ($p = 0.01$, Fig. 2b). We hypothesized that antibiotic usage did not directly impact HIV-1 infection but instead, influenced levels of the biomarkers (cytokines and gut microbiome). Consequently, instead of treating antibiotic usage as a confounder, we treated it as a covariate in the subsequent analyses between the sexual exposure groups and biomarkers. Both HBV infection and history of STI positively correlated with HIV-1 infection but were not correlated with the number of partners with whom a participant had receptive anal intercourse (Fig. 2). Hence, rather than being confounders in this study, they could serve as exposures in a different causal pathway leading to HIV-1 infection.

## Association analyses

To evaluate whether changes in biomarker levels, namely, inflammatory cytokines, and the gut microbiome mediate the effects of sexual behavior on HIV-1 infection, we first examined the associations between levels of these biomarkers with both exposure and outcome. Notably, our previously published work[21] detailed associations between each of these biomarkers and HIV-1 infection status. In this current study, we (a) investigated the association between the biomarkers, namely, cytokines, and microbiome, and the sexual exposure groups, and (b) reanalyzed the association between the microbiome and HIV-1 infection status using ANCOM-BC2[24], a recently developed methodology that has better performance characteristics

### Sexual exposure groups and plasma inflammatory cytokine levels.
For each participant, we assessed plasma for sCD14, soluble scavenger receptor CD163 (sCD163), interleukin 6 (IL-6), and lipopolysaccharide-binding protein (LBP) to detect monotonic increasing trends with an increase in the number of partners with whom a participant had receptive anal intercourse. As shown in Fig. 3a–c, there was a significant increasing trend in sCD14 levels with sexual exposure groups ($p = 0.014$), and marginally significant increasing trends in C-reactive protein (CRP) ($p = 0.083$) and sCD163 ($p = 0.068$) with sexual activity groups.

### Sexual exposure groups and gut microbiome.
A total of 13,073,544 sequence reads were generated for the 243 stool samples, with a median of 51,125 (range 67–126,903) reads per stool sample. The analyses did not demonstrate significant associations between alpha diversity metrics,

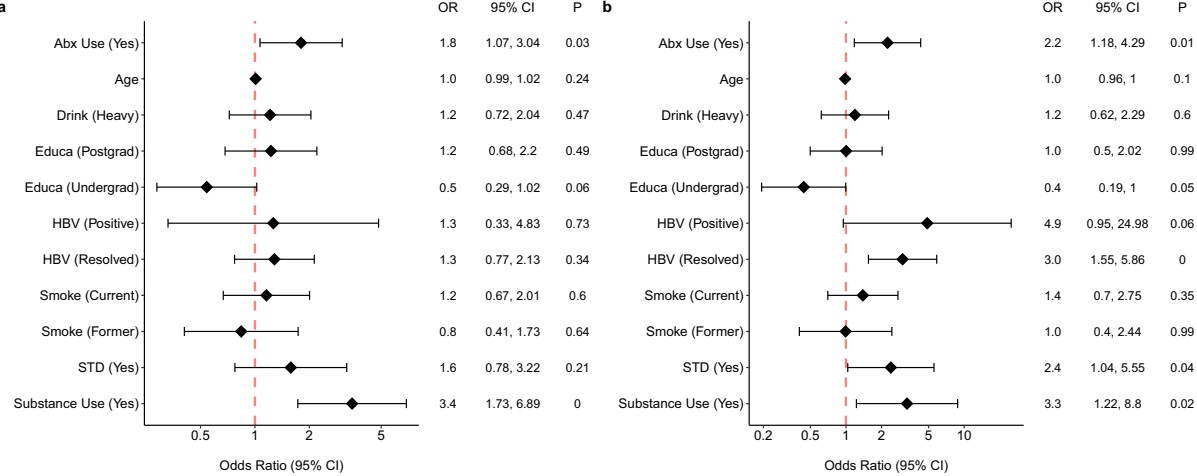

**Fig. 2 | Forest plot depicting the associations of demographic, clinical, and behavior features with exposures and the outcome.** This forest plot visualizes the associations of participants' features ($N = 241$) with **a** sexual exposure groups, analyzed using ordinal logistic regression models, and **b** the HIV-1 infection status outcome, analyzed using logistic regression models. The $X$-axis denotes the odds ratio (OR), while the $Y$-axis lists the various demographic and clinical

characteristics. Each feature's effect size (OR) is symbolized by a diamond. An accompanying horizontal line represents the 95% confidence interval (CI), indicating the range in which the actual effect size is likely to reside. A vertical red line at the OR of 1.0 serves as a reference for no effect. For each feature, the exact OR, 95% CI, and the $p$-value (two-sided)—derived from logistic regression models—are presented in an adjacent table.

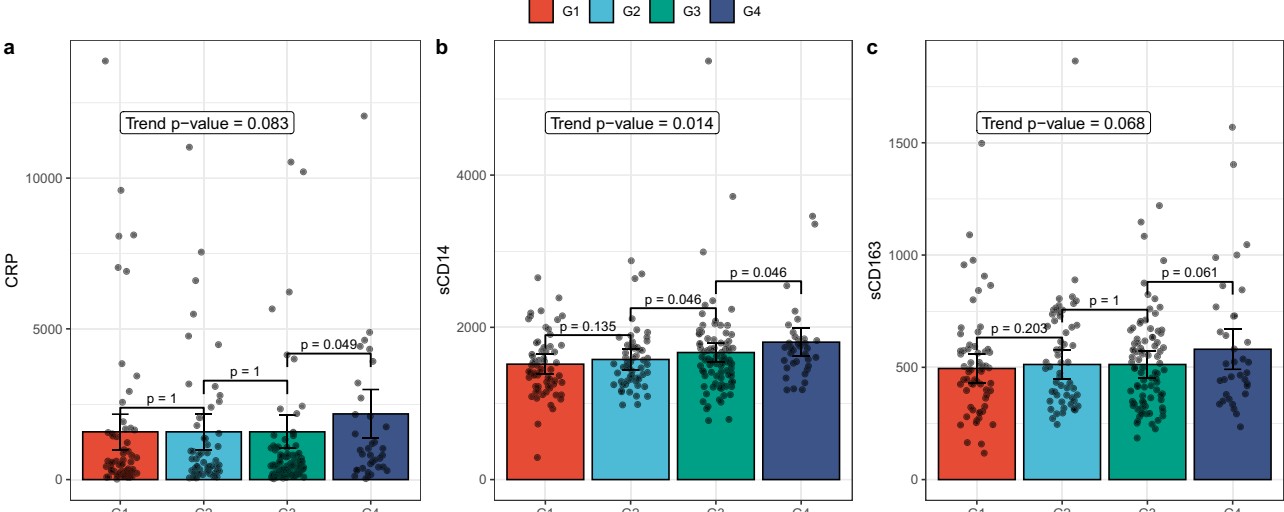

**Fig. 3 | Bar plots demonstrating monotonically increasing trends between exposure groups and cytokines levels.** Effect sizes of **a** CRP, **b** sCD14, and **c** sCD163 levels from participants' plasma samples ($N = 241$) were shown here. The *X*-axis details the exposure groups of participants ($N = 241$), from G1 to G4. The *Y*-axis reflects the cytokine's effect size determined by the CLME model. It is important to note that these are not raw concentrations but fitted values under a monotonic trend. Each bar's error bars denote the 95% CI. Pairwise *p*-values (one-sided), used for contrasting the exposure groups, are displayed above the brackets that span the respective bars. The *p*-value for a monotonically increasing trend is provided within the plot.

including richness and the Shannon diversity index, and the ordered sexual exposure groups. This held true across both an increasing trend (Supplementary Fig. 2a, b) and a decreasing trend (Supplementary Fig. 2c, d). Furthermore, using the Bray–Curtis dissimilarity, no discernible differences were observed in Beta diversity across sexual activity groups at the species level (Supplementary Fig. 3).

The ANCOM-BC2 pattern analysis was employed to evaluate monotonic increasing or decreasing trends in abundances of microbial species with sexual exposure groups (Fig. 4a). As the number of partners with whom a participant has receptive anal intercourse increased from Group 1 to Group 4, we discovered a significant decreasing trend ($p < 0.05$) in the abundance of some of the well-known commensal bacteria and those involved in the production of short-chain fatty acids such as *A. muciniphila* ($p = 0.001$), *B. uniformis* ($p < 0.001$), *Bacteroides* spp. ($p < 0.001$), *Bifidobacterium* spp. ($p < 0.001$), *A. onderdonkii* ($p = 0.007$), *Anaerovibrio* spp. ($p = 0.009$), *B. adolescentis* ($p = 0.021$), *B. caccae* ($p = 0.03$), *B. fragilis* ($p = 0.026$), *Butyricimonas* spp. ($p = 0.034$), *Lachnobacterium* spp. ($p = 0.029$), *Lachnospira* spp. ($p = 0.006$), *Megasphaera* spp. ($p = 0.006$), *Odoribacter* ssp. ($p = 0.015$), *Paraprevotella* spp. ($p = 0.02$), and *Succinivibrio* spp. ($p = 0.024$). The decreasing trends in *B. uniformis*, *Bacteroides* spp., and *Bifidobacterium* spp. were significant even after performing multiple testing *p*-value corrections, and the significance of *Bifidobacterium* spp. is robust to the presence of zero counts based on the sensitivity analysis. On the other hand, there was a significant increasing trend ($p < 0.05$) in the abundance of *C. celatum* ($p = 0.018$), *Dehalobacterium spp.* ($p = 0.013$), *Methanobrevibacter* spp. ($p = 0.016$), and *RFN20* spp. ($p = 0.021$) from Group 1 to Group 4 (Fig. 4a).

We implemented an ANCOM-BC2 two-group comparison to discern species exhibiting differential abundance (DA) in relation to HIV-1 infection status. Notably, among the species identified as significantly differentially abundant with regards to HIV-1 infection status (Fig. 4b), the following species also displayed monotonic trends with sexual exposure group: *A. muciniphila* ($p = 0.043$), *B. caccae* ($p = 0.043$), *B. fragilis* ($p < 0.001$), *B. uniformis* ($p = 0.006$), *Bacteroides* spp. ($p < 0.001$), *Butyricimonas* spp. ($p = 0.001$), *Dehalobacterium* spp. ($p < 0.001$), *Methanobrevibacter* spp. ($p = 0.006$), and *Odoribacter* spp. ($p = 0.013$). The effect sizes in *B. caccae* and *Dehalobacterium* spp. remained significant after performing multiple testing *p*-value corrections. Moreover, the significance of *B. caccae* proved to be robust to zero counts, as verified by our sensitivity

analysis. Interestingly, both *Dehalobacterium* spp. and *Methanobrevibacter* spp. potentially increased in abundance in future infection relative to uninfected participants, while the remaining species decreased in abundance among HIV-1 infected relative to uninfected participants.

**Interactions between microbial abundances and plasma cytokine levels that were associated with sexual exposure groups.** The differences in levels of inflammatory markers CRP, sCD14, and sCD163, previously established as significantly associated with the exposure variable (number of receptive anal intercourse partners), were not found to be significantly associated with the DA species among the sexual exposure groups, with two exceptions noted within the genera *Lachnospira* and *RFN20* (Supplementary Table 1).

**Mediation analysis**

The primary aim of this study was to assess whether inflammatory cytokines and gut microbiota mediate the relationship between an increase in the number of partners for receptive anal intercourse and subsequent HIV-1 infection (refer to Supplementary Fig. 4a). We focused on specific biomarkers including cytokines sCD14 and sCD163, and DA species *A. muciniphila*, *B. caccae*, *B. fragilis*, *B. uniformis*, *Bacteroides* spp., *Butyricimonas* spp., *Dehalobacterium* spp., *Methanobrevibacter* spp., and *Odoribacter* spp. These biomarkers were selected since they were identified as having significant associations with both sexual exposure groups, as well as the outcome variable, HIV-1 infection status.

Using the natural effect model consisting of sexual exposure groups as the exposure variable, biomarkers (sCD14 and sCD163) as the mediators, and HIV-1 infection status as the outcome variable, we discovered a significant natural direct effect (NDE) of the exposure on the outcome while controlling for biomarker levels. NDE quantifies the effect of the exposure on the outcome not through the mediator. Specifically, conditional on substance use, increasing the exposure from Group 1 to another group (while maintaining sCD14 and sCD163 at the same level) significantly increased the odds of HIV-1 infection. The odds ratios for Groups 2, 3, and 4 are $\exp(1.92) = 6.82$, $\exp(2.56) = 12.94$, and $\exp(3.55) = 34.81$, respectively (Table 2). This increasing trend is highly significant with *p*-value $< 0.001$ (Table 2)[25]. In addition to NDE, we also investigated the natural indirect effect (NIE) of the exposure on the outcome that acts through the mediator. Unlike NDE, NIE represents the effect of the intervention on the outcome

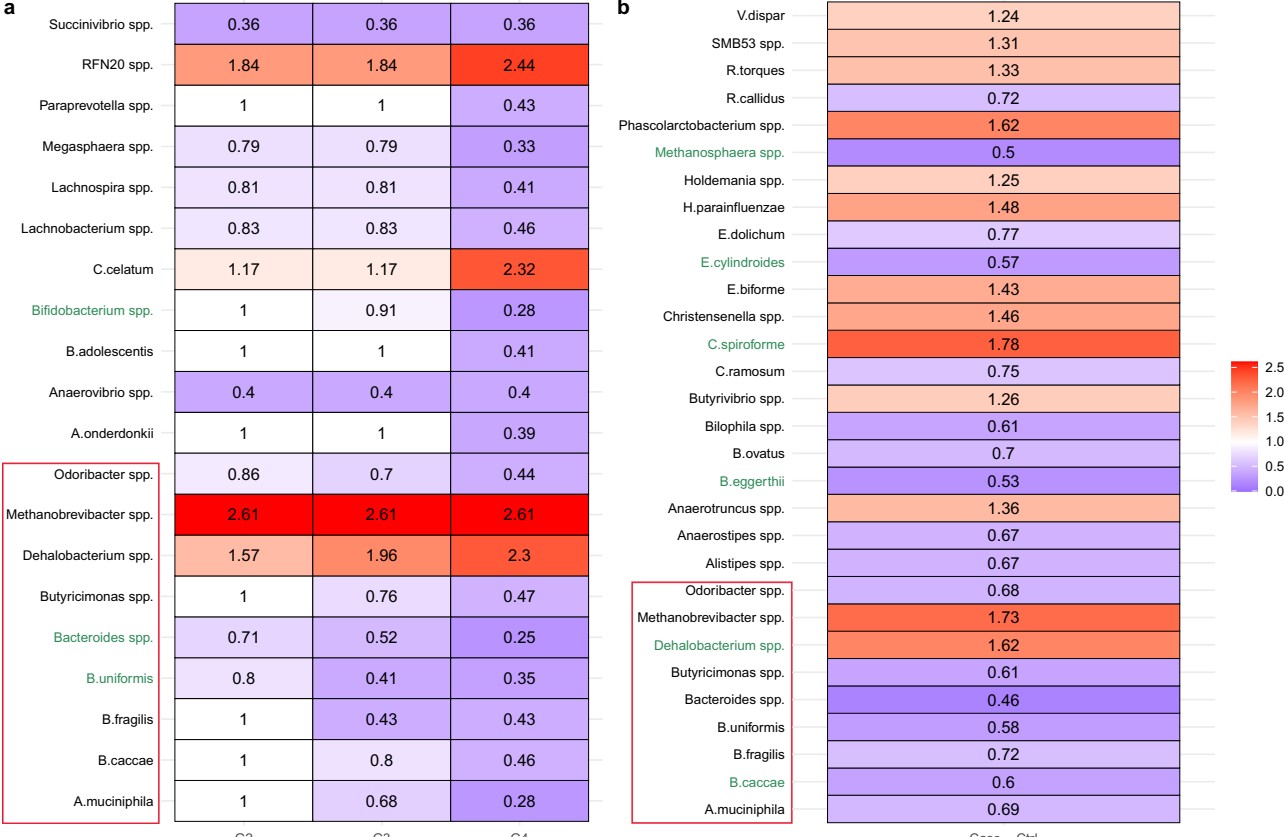

**Fig. 4 | Heatmaps depict the ANCOM-BC2 pattern analysis.** Monotonic increasing and decreasing trends in microbial species abundances from participants' stool samples (N = 241) were assessed in relation to **a** exposure groups (G1 is the reference) and **b** outcome of HIV-1 infection status. The X-axis delineates contrasts between exposure groups or outcomes, while the Y-axis lists species identified as significant via ANCOM-BC2 pattern analysis. Species that remained significant post-adjustment for multiple comparisons using the Benjamini–Yekutieli (BY) are highlighted in green. Fold-changes (natural log base) are superimposed within each cell. The color spectrum, from blue to red with a neutral white midpoint, visualizes the fold changes. Specifically, blue cells indicate reduced abundance relative to the reference group, and red cells signal increased abundance compared to the reference group. A white cell represents no effect, where the fold change equals 1. Species that are significantly associated with both sexual exposure groups, as well as HIV-1 infection status are highlighted in the red boxes.

solely due to its effect on the mediator. The natural effect models indicated a significant NIE of sCD14 and sCD163 levels on HIV-1 infection while controlling for the exposure among participants who had the largest number of receptive anal intercourse partners (p = 0.02, Table 2). A trend analysis further strengthened this result with a significant increase (p = 0.007, Table 2) in NIE corresponding with biomarker levels for participants that were exposed to a higher number of partners for receptive anal intercourse. Specifically, conditional on substance use, shifting levels of sCD14 and sCD163 from those observed in Group 1 to those seen in Group 4, while holding the exposure constant at any given group, increases the odds of HIV-1 infection with an odds ratio of exp(0.33) = 1.39.

A similar natural effect model consisting of sexual exposure groups as the exposure variable, microbiota *A. muciniphila, B. caccae, B. fragilis, B. uniformis, Bacteroides* spp., *Butyricimonas* spp., *Dehalobacterium* spp., *Methanobrevibacter spp.,* and *Odoribacter* spp. as mediators, and HIV-1 infection status as the outcome variable, we discovered a significant NDE of the exposure on the outcome while controlling the microbial abundances. Conditional on substance use, changing the exposure from Group 1 to any other group, while maintaining microbial abundance, increased the odds of HIV-1 infection (odds ratios for Groups 2, 3, and 4 are exp(2.08) = 8.00, exp(2.61) = 13.60, and exp(3.58) = 35.87, respectively; Table 2). A trend analysis further strengthened this result with a significant increasing trend (p < 0.001, Table 2). Moreover, this model highlighted a significant NIE of microbial abundances in HIV-1 infection, particularly among participants with the largest number of receptive anal intercourse partners (p = 0.04, Table 2). Trend analysis further validated a notable increase in NIE

associated with microbial abundance for participants that were exposed to a greater number of receptive anal intercourse partners, with a p-value of 0.033 (Table 2). Specifically, transitioning microbial species abundances from those of Group 1 to those in Group 4, while holding exposure constant, increased HIV-1 infection odds by exp(0.35) = 1.42.

When the cytokines and microbial species were taken together as mediators into the natural effect models, which is biologically reasonable to consider, the results of NDE for this combined model were similar to NDE results when separate natural models were considered for cytokines and microbial species (Table 2). However, there was a substantial increase in the NIE estimates from the combined model consisting of cytokines and microbiomes as mediators (Table 2) when compared to the two sets of mediators, cytokines, and microbiomes, modeled separately. For example, the NIE estimates of cytokines and microbiomes in Group 4 relative to Group 1, was about 1.5 times the NIE estimates of either cytokines or microbiomes in Group 4 relative to Group 1, when modeled separately. Thus, the microbiome and the cytokines synergistically mediated the effects of sexual exposure on HIV-1 infection. Interestingly, sexual exposure groups had significant NDE on the outcome through many microbial species and cytokines when taken individually (i.e., separate models for each of these variables) (Supplementary Table 2). However, individually, almost none of these variables demonstrated significant NIE mediating the effect of sexual activity on HIV-1 infection (Supplementary Table 2). We also investigated if substance use could be a potential mediator of sexual exposure, as defined in this paper, on HIV-1 infection and found it not significant (data not presented). Taken together, the above results suggest that

**Table 2 | Results of the natural effect models consisting of sexual exposure groups as the exposure variable, biomarkers (cytokines, microbial species, or their combined effects) as the mediators, and HIV-1 infection status as the outcome variable**

| NDE | | | | | |
|---|---|---|---|---|---|
| **Mediator** | **Comparison** | **LOR** | **SE** | *p*-value | **Trend test *p*-value** |
| | G2–G1 | 1.92 | 0.64 | 0.003 | <0.001 |
| sCD14, sCD163 | G3–G1 | 2.56 | 0.61 | 0.001 | |
| | G4–G1 | 3.55 | 0.7 | 0.001 | |
| | G2–G1 | 2.08 | 0.68 | 0.002 | <0.001 |
| *A. muciniphila, B. caccae, B. fragilis, B. uniformis, Bacteroides* spp., *Butyricimonas* spp., *Dehalobacterium* spp., *Methanobrevibacter* spp., *Odoribacter* spp. | G3–G1 | 2.61 | 0.64 | 0.001 | |
| | G4–G1 | 3.58 | 0.71 | 0.001 | |
| | G2–G1 | 1.81 | 0.64 | 0.005 | <0.001 |
| sCD14, sCD163, *A. muciniphila, B. caccae, B. fragilis, B. uniformis, Bacteroides* spp., *Butyricimonas* spp., *Dehalobacterium* spp., *Methanobrevibacter* spp., *Odoribacter* spp. | G3–G1 | 2.38 | 0.61 | 0.001 | |
| | G4–G1 | 3.16 | 0.68 | 0.001 | |
| **Natural indirect effect (NIE)** | | | | | |
| | G2–G1 | 0.11 | 0.1 | 0.284 | 0.007 |
| sCD14, sCD163 | G3–G1 | 0.12 | 0.09 | 0.168 | |
| | G4–G1 | 0.33 | 0.14 | 0.023 | |
| | G2–G1 | 0.02 | 0.13 | 0.879 | 0.033 |
| *A. muciniphila, B. caccae, B. fragilis, B. uniformis, Bacteroides* spp., *Butyricimonas* spp., *Dehalobacterium* spp., *Methanobrevibacter spp., Odoribacter* spp. | G3–G1 | 0.15 | 0.13 | 0.264 | |
| | G4–G1 | 0.35 | 0.17 | 0.045 | |
| | G2–G1 | 0.2 | 0.17 | 0.241 | 0.001 |
| sCD14, sCD163, *A. muciniphila, B. caccae, B. fragilis, B. uniformis, Bacteroides* spp., *Butyricimonas* spp., *Dehalobacterium* spp., *Methanobrevibacter* spp., *Odoribacter* spp. | G3–G1 | 0.29 | 0.17 | 0.087 | |
| | G4–G1 | 0.74 | 0.24 | 0.002 | |

Trend test *p*-values were derived using the methodology detailed in ref. [24].
*LOR* log odds ratio (natural log base).

collectively rather than individually the biomarkers sCD14 and sCD163, and the microbial species *A. muciniphila, B. caccae, B. fragilis, B. uniformis, Bacteroides* spp., *Butyricimonas* spp., *Dehalobacterium* spp., *Methanobrevibacter* spp., and *Odoribacter* spp. mediate the effects of sexual behavior on HIV-1 infection.

## Discussion

Kingsley et al. [7] previously reported receptive anal intercourse frequency (or number of partners) in the MACS was positively associated with HIV-1 infection because it increased the odds of exposure to the virus. The results of the present study reveal an intricate network of sexual behavior, immune response, and microbiota composition that greatly impact one's susceptibility to HIV-1 infection. Importantly, we show that sexual behaviors of MSM drive the gut microbiome dysbiosis and increased levels of plasma inflammatory cytokines observed 3 months prior to their becoming infected with HIV-1[21], indicating more inflammation of local rectal mucosal tissues, which were more vulnerable to HIV-1 infection even upon the same level of virus exposure.

It was striking to see a reduction in the abundance of several species of the genera Bacteroides in the higher sexual exposure groups compared to men in the no exposure group (Group 1), i.e., those who did not have receptive anal intercourse (Fig. 4a). These species were also reduced in abundance in HIV-1 infected compared to uninfected participants (Fig. 4b). Many species of *Bacteroides* are commensal gut bacteria and play a prominent role in gut health[26]. For example, *B. fragilis* whose abundance was reduced in higher sexual exposure groups compared to men in the no exposure group (Group 1) and reduced in the HIV-1 infected group compared to the uninfected group, plays an important role in preventing the expansion of infected Th17 cells and protects from potential damage to

mucosal barrier. Similarly, *B. uniformis*, which has reduced abundance in both high exposure groups, as well as the HIV-1 infected group, is involved in the production of IL-10, TNF-alpha, and in the protection against metabolic and immunological dysfunction[27], and is thus considered to be of therapeutic importance[28]. Similar to various species of *Bacteroides*, there is a reduction in *A. muciniphila* in the higher sexual exposure groups compared to men in the no-exposure group (Group 1), i.e., those who did not engage in receptive anal intercourse (Fig. 4a). This species was also reduced in abundance in HIV-1 infected group compared to the uninfected group (Fig. 4b). This species is known to be involved in numerous functions such as mucin degradation which is necessary to produce acetate and propionate, forming a substrate for other bacteria and host[29]. It is also involved in tissue repair of the intestinal mucosa and is thus important for the integrity of the intestinal mucosa[30], is anti-inflammatory, involved in immunomodulation, and is involved in the regulation of body weight.

Interestingly, pairwise correlations among almost all these taxa decreased over the sexual exposure groups. Specifically, the estimated correlations in the highest sexual group (Group 4) were zero for many taxa (Supplementary Fig. 5). These findings, together with reduction in the abundance of *A. muciniphila, B. caccae, B. fragilis, B. uniformis, Bacteroides* spp., *Butyricimonas* spp., and *Odoribacter* spp., and a potential increase in the abundance of low abundance taxa *Dehalobacterium* spp. and *Methanobrevibacter* spp., possibly point to gut dysbiosis with an increase in the number of partners with whom a participant had receptive anal intercourse leading to HIV-1 infection.

Our mediation analysis, employing natural effect models, illuminates the intricate interplay of sexual behavior, immune response, and gut microbiota in relation to HIV-1 infection. We found that the number of receptive anal intercourse partners stands as a significant predictor of HIV-1

infection, with remarkably large effect sizes for NDEs. Further, our study reveals significant natural indirect effects of two cytokines (sCD14 and sCD163) and multiple microbial species (*A. muciniphila*, *B. caccae*, *B. fragilis*, *B. uniformis*, *Bacteroides* spp., *Butyricimonas* spp., *Dehalobacterium* spp., *Methanobrevibacter* spp., *Odoribacter* spp.) in men who engage in receptive anal intercourse with a large number of partners. This finding highlights the potential role of these biological markers as mediators in the relationship between high-risk sexual behaviors and HIV-1 infection. Notably, these significant indirect effects were observed primarily among individuals engaging in more extreme risky sexual behaviors, such as having many receptive anal intercourse partners. Moreover, although we considered a simplified causal pathway described in Supplementary Fig. 4a to serve as an exploratory mediation analysis for screening potential mediators, a more comprehensive causal pathway from risky sexual behavior to HIV-1 infection described in Supplementary Fig. 4b needs to be considered. However, to perform such an analysis, more sophisticated methods are necessary which are beyond the scope of this study. While we find statistically significant results for *Dehalobacterium* spp. and *Methanibrevibacter* spp., due to the relatively low abundance of these taxa these findings need to be interpreted cautiously.

In addition to considering the number of receptive anal intercourse partners as the exposure, we probed the status of other STIs and HBV as alternative exposures with respect to the outcome of HIV-1 infection. Our results revealed that STI status showed a marginally significant association with IL-6 ($p = 0.05$). Nevertheless, no significant DA of microbial genera or species was observed in relation to STI status. On the other hand, HBV status (resolved vs. negative, positive group was excluded due to a small sample size) was marginally significantly associated with IL-6 ($p = 0.05$), significantly associated with CRP ($p = 0.01$), and significantly linked to DAs of the microbial genus *Alistipes* ($p < 0.001$) and two microbial species, *Parabacteroides stercorea* and *Alistipes putredinis* (both $p < 0.001$). Utilizing the natural effect models, we noted that while considering STI history, HIV-1 infection, and biomarker mediators, a statistically significant NDE was discerned. However, the natural indirect effect was not found to be significant, suggesting that STIs play a pivotal role in the pathway to HIV-1 infection. This effect does not appear to be mediated by the levels of cytokines or the gut microbiota. Lastly, when implementing the natural effect models while considering HBV, HIV-1 infection, and biomarker mediators, we identified a statistically significant, NDE, as well as a natural indirect effect for the microbial mediator *Alistipes*. This finding accentuates the crucial role of microbiome biomarkers on HIV-1 infection.

These insights underscore the complexity of the factors contributing to HIV-1 infection, demonstrating the interconnectedness of sexual behavior, immune response, and microbiota composition, especially among MSM participating in high-risk sexual behaviors[31]. Based on the principles of mediation analysis we discovered that the gut microbiome together with pro-inflammatory cytokines mediate the effects of sexual activity on HIV-1 infection. Additional studies would be necessary to confirm our findings.

We recognize the limitations in our study of the gut microbiome with a single stool specimen per participant. These include day-to-day variation and sample bias, which could impact the accuracy and reproducibility of the findings. Nevertheless, we believe that these data from our 40-year-old, single stool samples provide unique, unprecedented insights into the effects of sexual activity on risk for HIV-1 infection in MSM. Notably, this study impacts our understanding of the role of the microbiome and immune response in the context of HIV-1 transmission in the mid-1980s prior to the current public health emphasis on safer sex, including the use of condoms, and pre-exposure prophylaxis (PrEP) and PEP (post-exposure prophylaxis) with antiretroviral drugs[3]. Our results are pertinent and timely regarding the contemporary risk for HIV-1 infection and STI[32] among MSM in that condomless sex has steadily increased in the last two decades[10,31]. Moreover, although PrEP is highly effective for preventing HIV-1 infection in MSM[33], PrEP has been associated with aberrations in gut microbiota[34,35]. In conclusion, our results emphasize the myriad disruption of gut microbiota in

MSM related to sexual activity and its significant contemporary consequences.

## Methods
### Study participants
A total of 109 HIV-1 infected participants (case) and 156 HIV-1 negative controls (ctrl) from the MACS were included in this study as described in ref. 21. The MACS was a prospective cohort study of HIV-1 infection in MSM established in 1983 at four sites (Baltimore, Maryland/Washington, DC; Chicago, Illinois; Los Angeles, California; Pittsburgh, Pennsylvania)[7,36,37], that combined with the Women's Interagency HIV Study (WIHS) in 2019 to form the MACS-WIHS Combined Cohort Study (MWCCS)[38]. MACS participants were studied at semiannual clinic visits with standardized interviews, physical examinations, and phlebotomy for laboratory testing, with cryostorage of plasma and serum and viable peripheral blood mononuclear cells. The study was conducted with institutional review board approval from all participating institutions, all ethical regulations relevant to human research participants were followed, and informed consent was obtained. Enrollment and clinical research of the MACS participants began April 1, 1984, with clinical research visits at 6-month intervals thereafter. During that early period of the AIDS epidemic, a substantial number of MACS participants acquired HIV-1 infection during study follow-up. Importantly, during this early time, HIV-1 serostatus could not be determined, as no HIV-1-specific diagnostic test was available. HIV-1 infection was subsequently determined with the stored blood plasma samples by enzyme-linked immunosorbent assay and confirmed by Western blot[37]. Similarly, plasma HIV-1 RNA loads were also retrospectively measured by quantitative reverse transcriptase polymerase chain reaction. The primary HIV-1 infection date was estimated as the midpoint between the last HIV-1 RNA negative and first HIV-1 RNA positive study visits.

### Collection of demographic, clinical, and behavioral data of participants
During each study visit, study participants provided demographic, clinical, and behavioral information. For the baseline visit, this information covered the past 2 years, whereas for all subsequent visits, it encompassed the period since their last MACS clinic visit (6 months). Participants were categorized into four ordered groups (Table 1) based on participant responses regarding the number of partners with whom a participant had receptive anal intercourse, the exposure variable of interest. The number of receptive anal intercourse ranges from 0 to 200, with a mean of 4 and a median of 2. The distribution of receptive anal intercourse is highly skewed, with some participants reporting extremely high numbers (see Supplementary Fig. 6), potentially influenced by reporting bias. Categorizing the exposure variable helps mitigate this bias. Therefore, we categorized the exposure variable into the following 4 groups. Group 1 (G1) had no receptive anal intercourse partners ($N = 63$); Group 2 (G2) had one receptive anal intercourse partner ($N = 57$); Group 3 (G3) had 2 to 5 receptive anal intercourse partners ($N = 86$); and Group 4 (G4) had 6 or more receptive anal intercourse partners ($N = 35$). For simplicity of exposition, throughout this paper, we shall use the phrase "sexual exposure groups" to mean the above four ordered groups.

### Stool and plasma samples
During this early phase of the HIV-1 epidemic in 1984–1985, MACS participants were instructed to provide stool, urine, semen, and oral wash samples at each clinic research visit; these were preserved at −80 °C without additives or preservatives[37]. For this study, the stool and plasma samples were obtained from specimen cryo-repositories of the MACS. Stools were self-collected in 20 ml screw-capped glass vials at home and delivered to the clinic within one day by the participants. In the current study, we examined stool and plasma samples from the study visits flanking the estimated HIV-1 infection time point from 109 HIV-1 infected MSMs, and from 156 HIV-1 uninfected, age and race-matched MSM participant controls collected

during the same time period at the same MACS center. Plasma HBV core antibody, HBVsAg, and HCV antibody (Ab) were tested retrospectively by commercially available ELISA. The paired samples spanned approximately 6 months, with the 3-month midpoint designated as the date of HIV-1 infection. For defining HIV-1 infection in MSMs, pre-HIV-1 infection visits (plasma HIV-1 antibody negative and HIV-1 RNA negative) and post-HIV-1 infection visits (with detectable HIV-1 RNA in plasma) were stringently selected. For the HIV-1 negative controls, both visits were confirmed for plasma HIV-1 antibody negative and HIV-1 RNA negative.

## Profiling microbial populations by sequencing of the variable region of the 16S rRNA gene

As described in our previous study[21], fecal DNA was extracted from the stool samples, the V4 variable region in the 16S rRNA gene was PCR-amplified with the universal primers: 515F 5'-(GTG CCA GCM GCC GCG GTA A)-3' and 806R 5'-(GGA CTA CHV GGG TWT CTA AT)-3'[39] and the amplicons were sequenced on an Illumina MiSeq platform. The datasets generated and/or analyzed during the current study are available in the GitHub repository[40].

## 16S rRNA gene sequence analysis

The resulting 16S rRNA gene sequence data were processed using QIIME2 (version 2019.10.0). The raw sequence data were first demultiplexed and then denoised to remove noisy reads and dereplicate sequence, and clustered into amplicon sequence variants (ASVs) using the DADA2 algorithm[41]. The observed counts of ASVs were organized into a large matrix referred to as the feature table, where columns represent samples and rows represent ASVs. No ASV was removed based on its observed abundance. The taxonomic composition of bacterial communities was investigated by classifying sequences to the latest reference database[42] using a Naive Bayes classifier.

## Measurement of plasma inflammatory cytokines

As described in our previous study[21], the plasma inflammatory cytokines sCD14, sCD163, CRP, interferon γ-induced protein 10 (IP-10), and LBP were measured with the Luminex xMAP platform (Luminex, Northbrook, IL, USA). In addition, the inflammatory cytokine IL-6 was measured in the plasma samples by ELISA using a commercially available ELISA kit (R&D, Minneapolis, MN, USA).

## Statistics and reproducibility

Unless otherwise noted, all analyses presented in this study were performed using data from the MACS clinic baseline visit 1. For each study participant, the data from one paired stool and plasma samples were used. A significance threshold was set at α = 0.05. Multiple testing corrections were not applied to the p-values unless explicitly specified. Also, unless stated otherwise, all analyses in this article were carried out using the default settings in the respective software packages.

## Marginal association between demographic and clinical features and exposure groups and the outcome.

Ordinal logistic regression models were applied to investigate the marginal associations between the ordered sexual exposure groups and individual demographic and clinical features. This analysis utilized the "polr" function from the "MASS" package in R[43]. Likewise, logistic regression models were employed to explore the marginal associations between the outcome (HIV infected vs uninfected participants) and individual demographic and clinical features. This analysis was conducted using the "glm" function from the "stats" package in R.

## Analysis of gut microbiota diversity, DA, and correlations.

Alpha (within-sample) and beta (between-sample) diversity indices were calculated utilizing the R "microbiome" package[44] applied to rarefied data at the species level. Specifically, species were subsampled without replacement, based on the 10th percentile of library sizes across all samples,

corresponding to a total of 12,158 species. For the alpha diversity, we considered both the Shannon diversity index, as well as species richness. Adjusting for bacterial antibiotics usage, trends over the ordered sexual exposure groups, and pairwise comparisons among the sexual exposure groups, all while, were performed using the "CLME" R package[45]. For the beta diversity, we used the Bray–Curtis dissimilarity measure, and the statistical significance was determined through pairwise Permutational Multivariate Analysis of Variance[46]. Bacterial species displaying monotonically increasing or decreasing trends over the four sexual exposure groups were identified by the Analysis of Compositions of Microbiomes with Bias Correction 2[47] without applying a sensitivity score filter. Bacterial antibiotics usage was adjusted as a covariate. Taxa with prevalence less than 10% (prv_cut = 0.1) and samples with library sizes less than 1000 (lib_cut = 1000) were excluded from the analysis. The p-values associated with these trends were estimated based on 1000 bootstrap samples. Additionally, ANCOM-BC2 without a sensitivity score filter was employed to identify DA species between HIV-1 infected and uninfected participants. Pairwise Pearson correlations between species were estimated using Sparse Estimation of Correlations among Microbiomes[48]. Species yielding raw p-values < 0.05 were shown in the corresponding ANCOM-BC2 results. Those that remained significant after adjustment for multiple comparisons using the Benjamini–Yekutieli procedure[49] are highlighted in the figure.

## Interactions between different biomarkers.

To investigate the multivariate relationships among cytokines, and the microbial species that were identified as associated with sexual exposure groups, the multivariate analysis of covariance (MANCOVA) using the Pillai–Bartlett trace statistic was performed[50]. To account for the compositionality of microbiome data, the taxon, and sample-specific bias-corrected log microbial abundances obtained from the ANCOM-BC2 model were utilized in this investigation.

## Mediation analyses.

To understand whether changes in biomarker levels (microbiome, and inflammatory cytokines) mediate the effect of sexual exposure groups on the outcome (HIV-1 infection), we employed natural effect models[51–54]. We focused on the potential mediating effects of biomarkers (cytokines, and the gut microbiome) that were significantly associated with the sexual exposure group, as well as the outcome variable, namely, HIV-1 infection status. We emphasize that, when analyzing microbial mediators, we accounted for the compositionality of microbiome data by using taxon and sample-specific bias-corrected log microbial abundances derived from the ANCOM-BC2 model, rather than relying on raw abundances. Based on the results of univariable analyses involving demographic, clinical, and behavior features, sexual exposure groups, and the HIV-1 infection status outcome (Fig. 2), we identified substance use as the confounding variable for inclusion in the mediation analyses. It is crucial to highlight that the analysis of high-dimensional and compositional mediators, such as microbiome compositions, is an active research area and presents significant challenges. While several methodologies have been proposed, including MarZIC[55], CCMM[56], and SparseMCMM[57], computational software is not yet available. As a result, in this study, we adopted a strategy to first reduce the dimensionality of microbiome data through DA analysis, focusing on microbes significantly associated with exposure and outcome. This approach allows us to apply traditional mediation methods designed for low-dimensional mediators. For our analysis, we utilized Natural Effect Models, which has established and accessible software. Three multiple-mediator mediation analyses were performed: (1) an analysis incorporating all microbial mediators, (2) an analysis incorporating all cytokine mediators, and (3) an analysis including both microbial and cytokine mediators simultaneously. These natural effect models were fitted using an imputation-based approach[54], which is carried out in two steps. Initially, a working model is used to handle missingness in the outcome by estimating the outcome mean. Subsequently, the natural effect model

is fitted to the imputed data. It is important to note that this two-step approach in natural effect models differs from the product method[58], which involves two regressions: one regression fits the outcome on the exposure, the mediator, and the covariates, while the other regression fits the mediator on the exposure and the covariates. We performed the above mediation analysis using the R package "medflex"[59]. Imputations for counterfactuals were performed using the function "neImpute" with the setting "robust" for "se" to ensure robust standard errors for the parameter estimates. Further details regarding the assumptions and modeling techniques used in the natural effect models are provided in the Supplementary Methods.

**Sensitivity analysis.** To corroborate the robustness of our principal categorization strategy, we undertook a sensitivity analysis utilizing an alternative parameterization, termed the "secondary group". Analogous to our study's primary delineation, this secondary group stratifies participants based on their number of partners involving receptive anal intercourse: Group 1 (no partners, $n = 63$), Group 2 (one partner, $n = 57$), Group 3 (2 to 3 partners, $n = 56$), Group 4 (4 to 8 partners, $n = 42$), and Group 5 (9 or more partners, $n = 23$). The results derived from this secondary stratification were congruent with our primary findings, reinforcing the validity and resilience of our principal categorization method. To avoid redundancy, we have chosen not to display these results separately.

## Data availability
The processed datasets utilized in the present study have been made accessible through our GitHub repository[40]. The raw data, including the microbiome sequencing data, are not publicly available but will be obtained upon request as per MWCCS policy (https://statepi.jhsph.edu/mwccs/).

## Code availability
The code utilized for all analyses in this manuscript is available in the associated GitHub repository[40], as well as in the corresponding Code Ocean capsule (https://doi.org/10.24433/CO.5535732.v1).

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

## Acknowledgements

The authors gratefully acknowledge the contributions of the study participants and the dedication of the staff at the MACS sites. The authors thank Adam Fitch and Barbara Methe from the Department of Medicine, University of Pittsburgh School of Medicine, for laboratory technical support. The contents of this publication are solely the responsibility of the authors and do not represent the official views of the National Institutes of Health (NIH). Data in this manuscript were collected by the MACS, which is now the MWCCS. This research was supported in part by N01-AI-32513, U01-HL-146208, and NIHS10OD023402 from the National Institutes of Health. This work was performed in part at the University of Pittsburgh and supported through the Rustbelt Center for AIDS Research (P30AI036219). Research done by HL and SDP was supported in part by funding from the NIEHS intramural program ZIA ES103390-01.

## Author contributions

H.L. contributed to the study design, statistical methodology, code development, statistical analyses, interpretation of results, and writing the manuscript. Y.C. and C.R.R. contributed to the study design, interpretation of results, and writing the manuscript. A.M. contributed to microbiome detection. S.D.P. contributed to the study design, statistical methodology, interpretation of the results, and writing the manuscript. G.A. and M.P. contributed to the analysis of sexual activity data. J.S., F.P., K.C., and T.B. provided data from their respective MACS sites and reviewed drafts of the manuscript.

## Funding

## Competing interests

The authors declare no competing interests.
