## [Peer Review File · Communications Biology]

Reviewers' comments:

Reviewer #1 (Remarks to the Author):

The authors have appropriately softened language that could be misinterpreted as establishing causal relationships.

Reviewer #2 (Remarks to the Author):

This manuscript by Lin and colleagues investigates whether effect of sexual behaviours on HIV infection is mediated via either gut microbiome or cytokines. The authors highlight that a set of both microbial species and a set of cytokines mediate the effect, however, mediating effects via individual cytokines or microbial species were not found.

Please find below some comments regarding especially the statistical part of the manuscript. Please note that I have been asked to review only the statistical/mediation part of the text and thus have not focused as much on other parts of the manuscript. In addition, I did not have access to the responses for the first round of reviewer comments, my apologies if my questions concern something that was already solved on the first round.

Major comments:

1. I am pleased to see that the authors aim for transparency by sharing their data and codes on GitHub. However, I think it would be important to also report some methodological details in the text in a more detailed and transparent manner. In the following comments I have raised some more detailed observations related to this point.
2. The authors acknowledge that analysis of high-dimensional compositional mediators is active research area and even though methods for assessing the mediating role of a high dimensional mediator have been published, no computational software is available. Thus, the authors have used the “medflex” package which allows performing Natural Effect Models. I have a few comments/notes regarding this statement and the mediation analysis.
 - a) Here, the mediator is not high-dimensional, as the authors only focus on the species that were significantly associated with both exposure and outcome.
 - b) Importantly, compositional data analysis models have been indeed recommended in

previous literature. For example, Gloor et al. (2017) have presented rationale for treating microbiome data sets as compositions.

Even though the authors refer to microbial mediators as compositions, based on the text I remained uncertain whether the observed microbial counts were indeed treated as compositions or e.g. analysed as counts. It would be very important to report how the microbial data were handled in the mediation analysis. When using compositional data analysis methods, the compositional counts are transformed into coordinates. Not only does this approach remove the issues with count data highlighted by Gloor et al., but it also avoids the unit-sum constraint issue that may arise when working with compositions scaled into proportions.

Even with no established statistical software, scaling the counts within a taxonomic level into proportions and then using a compositional transformation on them should be a rather straightforward. I surmise that calculating centered or additive log-ratio transformations (clr and alr) could even be coded manually, while e.g. the philr package in R might help in building isometric log-ratio transformations (ilr). Presumably some further guidance could be found in the references 56 and 57 of the manuscript. I encourage the authors to have a look at these methods and try whether the compositional data analysis methods yield similar results as the current analyses.

c) Based on the results section, I assume that there were two primary natural effects models: one for all microbial mediators simultaneously, and one for all cytokine mediators simultaneously, i.e., there were two multiple-mediator mediation analyses? I think this could be stated more clearly in the methods, especially as the additional analyses focus on each mediator separately. In addition, it would be great to have a little bit more information on the multiple mediators and assumptions related to them. For example, can the mediators be correlated under this analytical framework, and were they correlated in this application? Does the assumption of “nonintertwined causal pathways” hold here? If so, it would be good to emphasise this in the text, too.

3. I think the authors have described in the results section the NDE and NIE clearly. In addition, the authors report the underlying theory in the supplementary material. It would be well appreciated if in the main text the authors would briefly describe the idea of the models. Would it be possible to also elaborate what are the key differences between these approaches and e.g. the “product-of-coefficients” method?

4. The authors report that “No ASV was removed based on its observed abundance”. Did

the authors choose not to remove any ASVs, or was there a criterion for removal that none of the ASVs exceeded (if so, what were the criteria)? It appears that the editor and other reviewers have in their previous comments raised the issue of data sparsity and low abundance of certain taxa. I agree with this concern, and while the authors assure that the differential abundance methods take sparsity into account, it is not clear for me whether this aspect has been considered in the mediation analyses. In addition, the lowest observed read per stool sample was 67. Have the authors considered a sensitivity analysis where the samples with very low read counts are removed?

Minor comments:

5. The results are reported in the text for “subject who had not used substances”, while 81% of the participants had reported past substance use. I see that substance use was identified as a confounder; does this mean that the analysis sample was stratified into substance users and non-users? Could the authors report the results similarly for the participants who had used substances?

6. In the DAGs a) and b) of the supplement, there are no arrows from the confounder (drug use) to the biomarkers (microbiome and cytokines, dysbiosis of microbiome). Is this on purpose?

7. Could the authors add a reference to the approach that was used to control for multiple testing in the methods section?

8. In supplementary material page 3, equation $\text{logit PR} = \dots$, there seems to be three β_{11} coefficients. Should the last two ones be β_{12} and β_{13} ?

9. On page 6, it states that “sexual exposure groups had significant NDE on many microbial species”. Could the authors elaborate what this means? Should it only read “significant effect” or “significant association” instead of NDE?

Response to reviewers' comments

We thank the reviewers for their helpful comments that improved the presentation of the manuscript substantially. In the following we provide item-by-item response (in plain text) to each comment we received (in italics).

Reviewer #1

The authors have appropriately softened language that could be misinterpreted as establishing causal relationships.

Our response: We thank the reviewer for all previous comments which have substantially improved the tone and presentation of the paper. We are happy to hear that this reviewer has no further comments and that we have addressed the concerns satisfactorily.

Reviewer #2:

This manuscript by Lin and colleagues investigates whether effect of sexual behaviours on HIV infection is mediated via either gut microbiome or cytokines. The authors highlight that a set of both microbial species and a set of cytokines mediate the effect, however, mediating effects via individual cytokines or microbial species were not found.

Please find below some comments regarding especially the statistical part of the manuscript. Please note that I have been asked to review only the statistical/mediation part of the text and thus have not focused as much on other parts of the manuscript. In addition, I did not have access to the responses for the first round of reviewer comments, my apologies if my questions concern something that was already solved on the first round.

Major comments:

1. I am pleased to see that the authors aim for transparency by sharing their data and codes on GitHub. However, I think it would be important to also report some methodological details in the text in a more detailed and transparent manner. In the following comments I have raised some more detailed observations related to this point.

Our response: Thank you. Please see below our responses to your comments.

2. The authors acknowledge that analysis of high-dimensional compositional mediators is active research area and even though methods for assessing the mediating role of a high dimensional mediator have been published, no computational software is available. Thus, the authors have used the “medflex” package which allows performing Natural Effect Models. I have a few comments/notes regarding this statement and the mediation analysis.

a) Here, the mediator is not high-dimensional, as the authors only focus on the species that were significantly associated with both exposure and outcome.

Our response: The reviewer is correct. While several mediation methods are available for microbiome data, they are often unsupported by reliable software, or the existing software contains bugs. Addressing these software issues is beyond the scope of this paper. Consequently, we adopted a strategy to first reduce the dimensionality of microbiome data through differential abundance analysis, focusing on microbes significantly associated with exposure and outcome. This approach allows us to apply traditional mediation methods designed for low-dimensional mediators. For our analysis, we chose the "medflex" package, which is well-suited for low-dimensional mediators and facilitates ease of interpretation. To provide further clarity, we have included additional explanations in the "Mediation analyses" section of the Methods part of our revised manuscript (see page 16), detailing our approach.

b) Importantly, compositional data analysis models have been indeed recommended in previous literature. For example, Gloor et al. (2017) have presented rationale for treating microbiome data sets as compositions.

Even though the authors refer to microbial mediators as compositions, based on the text I remained uncertain whether the observed microbial counts were indeed treated as compositions or e.g. analysed as counts. It would be very important to report how the microbial data were handled in the mediation analysis. When using compositional data analysis methods, the compositional counts are transformed into coordinates. Not only does this approach remove the issues with count data highlighted by Gloor et al., but it also avoids the unit-sum constraint issue that may arise when working with compositions scaled into proportions.

Our response: We totally agree with the reviewer. This is exactly the reason over the years we have been developing and promoting methods that address the underlying compositionality in the microbiome data (e.g., ANCOM, ANCOM-II, ANCOM-BC, SECOM, ANCOM-BC2)). Accordingly, in this paper we used ANCOM-BC2 to identify potential microbial mediators which respects the underlying compositionality in the data. Thus, consistent with the reviewer's expectations, all analyses reported in the paper regarding the microbiome account for compositionality. To enhance clarity, we have added the following sentence in the "Mediation analyses" paragraph in the Methods section: "We emphasize that, when analyzing microbial mediators, we accounted for the compositionality of microbiome data by using taxon and sample-specific bias-corrected log microbial abundances derived from the ANCOM-BC2 model, rather than relying on raw abundances" (page 15).

Even with no established statistical software, scaling the counts within a taxonomic level into proportions and then using a compositional transformation on them should be a rather straightforward. I surmise that calculating centered or additive log-ratio transformations (clr and alr) could even be coded manually, while e.g. the philr package in R might help in building isometric log-ratio transformations (ilr).

Presumably some further guidance could be found in the references 56 and 57 of the manuscript. I encourage the authors to have a look at these methods and try whether the compositional data analysis methods yield similar results as the current analyses.

Our response: Thank you for your suggestion. Consistent with our above response, we utilized the ANCOM-BC2 methodology to identify potential microbial mediators. To appropriately account for the compositionality of the data, we used bias-corrected log abundances derived from the ANCOM-BC2 model, rather than raw abundances. These bias-corrected log abundances conceptually resemble clr-transformed abundances, as ANCOM-BC2 estimates sampling fractions that vary across samples and subsequently models the log abundance through a linear regression model, incorporating the estimated sampling fraction as an offset term.

c) Based on the results section, I assume that there were two primary natural effects models: one for all microbial mediators simultaneously, and one for all cytokine mediators simultaneously, i.e., there were two multiple-mediator mediation analyses? I think this could be stated more clearly in the methods, especially as the additional analyses focus on each mediator separately. In addition, it would be great to have a little bit more information on the multiple mediators and assumptions related to them. For example, can the mediators be correlated under this analytical framework, and were they correlated in this application? Does the assumption of “nonintertwined causal pathways” hold here? If so, it would be good to emphasise this in the text, too.

Our response: The reviewer is correct. We implemented two primary natural effect models to explore the roles of both microbial and cytokine mediators separately. Additionally, we conducted an analysis that combined these mediators. To clarify this methodology, we have added the following sentence to the Methods section: “Three multiple-mediator mediation analyses were performed: 1) an analysis incorporating all microbial mediators, 2) an analysis incorporating all cytokine mediators, and 3) an analysis including both microbial and cytokine mediators simultaneously” (page 16). The basic assumptions required for mediation analysis, as outlined in VanderWeele, T. J. (2016) and Steen, J. et al. (2017), were also adhered to in our application of natural effect models. Specifically, these assumptions include: 1) the control of exposure-outcome confounding, 2) the control of mediator-outcome confounding, 3) the control of exposure-mediator confounding, and 4) the condition that there should be no mediator-outcome confounder affected by the exposure. In the context of natural effects models with multiple mediators, it employs a joint mediation approach that decomposes the total causal effect into components: one transmitted through the combined influence of multiple mediators and another that is independent of the mediators under consideration. We have included detailed descriptions of assumptions in the natural effect models in the Supplementary Methods (page 3) to provide clearer explanations and enhance understanding. We thank you for the suggestion.

3. I think the authors have described in the results section the NDE and NIE clearly. In addition, the authors report the underlying theory in the supplementary material. It would be well appreciated if in the main text the authors would briefly describe the idea of the models. Would it be possible to also

elaborate what are the key differences between these approaches and e.g. the “product-of-coefficients” method?

Our response: We appreciate the reviewer's suggestion and have revised the "Mediation analyses" section of the Methods (page 16) accordingly to improve clarity, and explicitly draw distinctions between the natural effect models and the product-of-coefficients method. We have added the following text: “These natural effect models were fitted using imputation-based approach, which is carried out in two steps. Initially, a working model is used to handle missingness in the outcome by estimating the outcome mean. Subsequently, the natural effect model is fitted to the imputed data. It is important to note that this two-step approach in natural effect models differs from the product method, which involves two regressions: one regression fits the outcome on the exposure, the mediator, and the covariates, while the other regression fits the mediator on the exposure and the covariates.” Additionally, we have expanded the Supplementary Methods section (page 4) to provide a more comprehensive explanation of the assumptions and the modeling strategy employed in the natural effect models.

4. The authors report that “No ASV was removed based on its observed abundance”. Did the authors choose not to remove any ASVs, or was there a criterion for removal that none of the ASVs exceeded (if so, what were the criteria)? It appears that the editor and other reviewers have in their previous comments raised the issue of data sparsity and low abundance of certain taxa. I agree with this concern, and while the authors assure that the differential abundance methods take sparsity into account, it is not clear for me whether this aspect has been considered in the mediation analyses. In addition, the lowest observed read per stool sample was 67. Have the authors considered a sensitivity analysis where the samples with very low read counts are removed?

Our response: We thank the reviewer for raising concerns regarding data sparsity. We wish to clarify that “No ASV was removed based on its observed abundance” from the sequencing data, indicating that no data were excluded prior to conducting the statistical analysis. However, for the differential abundance analysis, we applied filters to both taxa and samples as per the default settings in the ANCOM-BC2. To provide clarity, we have added the following sentence to the “Analysis of gut microbiota diversity, differential abundance (DA), and correlations” paragraph in the Methods section (page 15): “Taxa with prevalence less than 10% ($prv_cut = 0.1$) and samples with library sizes less than 1000 ($lib_cut = 1000$) were excluded from the analysis.”

Minor comments:

5. The results are reported in the text for “subject who had not used substances”, while 81% of the participants had reported past substance use. I see that substance use was identified as a confounder; does this mean that the analysis sample was stratified into substance users and non-users? Could the authors report the results similarly for the participants who had used substances?

Our response: Thank you for highlighting this. We did not conduct a stratified analysis, and as detailed in the Supplementary Methods, effect modifications in the baseline covariates (i.e., substance use) were

not considered. The phrase “subject who had not used substances” was initially used to indicate that the natural effects are conditional effects based on baseline covariates. For greater clarity, we have revised this to “conditional on substance use” instead in the Results section.

6. In the DAGs a) and b) of the supplement, there are no arrows from the confounder (drug use) to the biomarkers (microbiome and cytokines, dysbiosis of microbiome). Is this on purpose?

Our response: Thanks for the comment. It was an oversight on our part. We have now included the arrows.

7. Could the authors add a reference to the approach that was used to control for multiple testing in the methods section?

Our response: We indicated in the “Statistical methods” section that “Multiple testing corrections were not applied to the p-values unless explicitly specified.” We only implemented multiple testing correction in microbial differential abundance analysis, where Benjamini–Yekutieli procedure was used.

8. In supplementary material page 3, equation logit PR... =, there seems to be three β_{11} coefficients. Should the last two ones be β_{12} and β_{13} ?

Our response: Thanks for catching the typo! We have now corrected the notations.

9. On page 6, it states that “sexual exposure groups had significant NDE on many microbial species”. Could the authors elaborate what this means? Should it only read “significant effect” or “significant association” instead of NDE?

Our response: Thank you for pointing this out. We have revised the sentence as follows (page 6): “Interestingly, sexual exposure groups had significant NDE on the outcome through many microbial species and cytokines when taken individually (i.e., separate models for each of these variables) (Supplementary Table 2). However, individually, almost none of these variables demonstrated significant NIE mediating the effect of sexual activity on HIV-1 infection (Supplementary Table 2).”

REVIEWERS' COMMENTS:

Reviewer #2 (Remarks to the Author):

I thank the authors for addressing my comments and providing comprehensive responses to them. I think it is now easier to follow the approaches the authors have undertaken in their analyses and I find the level of detail sufficient.